# An experimental approach to training mood for resilience

**Vasileios Mantas**[1,2☯]*, **Vasileia Kotoula**[3☯], **Charles Zheng**[4], **Dylan M. Nielson**[4], **Argyris Stringaris**[1,5]*

**1** 1st Department of Psychiatry, National and Kapodistrian University of Athens, Aiginiteion Hospital, Athens, Greece, **2** National Institute of Mental Health, National Institutes of Health, Bethesda, Maryland, United States of America, **3** Experimental Therapeutics and Pathophysiology Branch, National Institute of Mental Health, National Institutes of Health, Bethesda, Maryland, United States of America, **4** Machine Learning Team, Section for Functional Imaging Methods, National Institute of Mental Health, National Institutes of Health, Bethesda, Maryland, United States of America, **5** Divisions of Psychiatry and Psychology and Language Sciences, University College London, London, United Kingdom

☯ These authors contributed equally to this work.
* a.stringaris@ucl.ac.uk (AS); mantas.vasilios@yahoo.com (VM)

**Data Availability Statement:** All data from this study are publicly available in the OSF repository (https://osf.io/ujyxa/).

**Funding:** This work was funded by the Intramural Research Program of the National Institute of

## Abstract

According to influential theories about mood, exposure to environments characterized by specific patterns of punishments and rewards could shape mood response to future stimuli. This raises the intriguing possibility that mood could be trained by exposure to controlled environments. The aim of the present study is to investigate experimental settings that increase resilience of mood to negative stimuli. For this study, a new task was developed where participants register their mood when rewards are added or subtracted from their score. The study was conducted online, using Amazon MTurk, and a total of N = 1287 participants were recruited for all three sets of experiments. In an exploratory experiment, sixteen different experimental task environments which are characterized by different mood-reward relationships, were tested. We identified six task environments that produce the greatest improvements in mood resilience to negative stimuli, as measured by decreased sensitivity to loss. In a next step, we isolated the two most effective task environments, from the previous set of experiments, and we replicated our results and tested mood's resilience to negative stimuli over time, in a novel sample. We found that the effects of the task environments on mood are detectable and remain significant after multiple task rounds (approximately two minutes) for an environment where good mood yielded maximum reward. These findings are a first step in our effort to better understand the mechanisms behind mood training and its potential clinical utility.

## Introduction

Mood is a defining component of subjective well-being [1], as the presence of positive emotions and moods and the absence of negative emotions are two of the core components of subjective well-being [2]. Moreover, mood disorders such as depression are common [3],

Mental Health, part of the National Institute of Health. The funder had no role in the design and conduct of the study; collection, management, analysis and interpretation of the data; preparation, review and approval of the manuscript; or decision to submit the manuscript for publication. The views expressed in this article do not necessarily represent the views of the National Institute of Health, the Department of Health and Human Services or the United States Government. The funders had no role in study design, data collection and analysis, decision to publish, or preparation of the manuscript.

**Competing interests:** The authors declare no conflicts of interest.

debilitating and potentially lethal [4]. Therefore, a great scientific effort has been invested in trying to understand how best to regulate and change mood through psychological and pharmacological therapies. In this paper, we use a different starting point, namely evolutionary theories about mood and create an experimental set-up to investigate whether mood could be trained to decrease its sensitivity to negative stimuli.

Mood is often considered a biological system that interacts with the environment for adaptive purposes. Under this perspective, the term 'mood' is used to describe relatively enduring affective states which arise when positive or negative experiences in one context or time period alter the individuals' threshold for responding to potential negative and positive stimuli [5]. This definition suggests that the interaction between environmental events and mood can alter the way mood will respond to future events. This, in turn, raises the possibility that by exposing individuals to specific environments we may be able to train their mood in such a way as to make it resilient to future environmental events.

The term resilience refers to the capacity of a dynamic system to withstand or recover from significant disturbances that threaten its adaptive function, variability or development [6]. In relation to mood and its interaction with the environment, mood resilience could be viewed as the ability of mood to resist the effects of exposure to adverse stimuli or bounce back from states provoked by such an exposure [7]. For the purposes of our study, and concerning resilience, we focus on the first component of the definition and explore whether mood could be trained to resist the effects of exposure to negative stimuli.

In order to explore whether mood could be trained to become resilient to negative stimuli, we used a well-established research framework of experiments which employ virtual environments and stimuli as a means to provoke mood changes [8–10]. The majority of these experiments study mood dynamics by using gambling tasks and study the effect of gains and losses on mood. These experiments have shown that momentary mood is affected by reward prediction errors which signify the difference between the expected and actual outcome of reward trials [11]. When examining the influence of recent and past experiences on mood, Keren et al. have shown that mood can be modeled as being informed by a weighted average of environmental events with the earliest experiences having a greater influence on the expectation of reward which drives momentary mood. Momentary mood has also been shown to be sensitive to the passage of time, as mood ratings during rest periods or even during a gambling task have been shown to spontaneously, drift downward [12].

For our experiments, we developed a new computerized task where participants were asked to rate their mood after points were added to or subtracted from their score. From the experimenter's perspective, points were used as positive and negative stimuli to change mood. Different relationships between mood ratings and immediate rewards defined different experimental environments. Since the task focused on mood training by the outcomes of task environments and not on the examination of the effects that different aspects of decision making (ex. reward prediction error) could have on mood, no action is required from the participants and stimuli are automatically administered. Such a setting allows one to examine the direct effect of environmental events (in this setting, task outcomes) on mood without any decision related effect such as effort required to perform the task actions, or regrets/satisfaction of previous decisions. Moreover, it could be paralleled to real-life events that impact our mood and do not require any action from the individual's behalf, including the weather, accidents, good and bad news.

We conducted a total of three sets of experiments.

The main contribution of our first set of experiments was mainly a methodological one, addressing a major, but strangely unexamined concern in the literature, namely that mood ratings in incentivised tasks may be merely reflections of the rewards received, rather than honest

mood ratings. To examine this, we hypothesized that the majority of participants would follow task instructions and register their true momentary mood and tested this by creating paradoxical reward environments, chiefly ones were participants were incentivized to rate negative moods.

Our second set of experiments was exploratory and focused on the identification of task environments that would promote mood resilience to negative stimuli. We hypothesized that environments where negative mood was punished or positive mood was rewarded would have a positive impact on mood's resilience to negative stimuli which we operationalize based on the impact of negative stimuli on mood ratings.

Finally, in the third set of experiments, we aimed to examine whether the training effect, if achieved in the previous set of experiments, would replicate and endure over time. We hypothesized that the positive effect of training on mood would still be present after 10 and 20 random stimuli.

## Methods

### General task design for all experiments

For this study, a task was developed with JavaScript using the PixiJS library. The task was executed using an internet browser and task responses were made either using the mouse or the keyboard. In an effort to engage participants, the task employed color animations. In this task, points were represented as coins, which were added to the participants' piggy bank when a reward was delivered. A magnet subtracted coins from the piggy bank every time a punishment was administered. At each task round, participants were asked to register their mood when points were added or removed from their piggy bank.

The task consisted of different experimental environments. Each environment was characterized by a specific relationship (for example proportional) between each mood rating, registered by the participants, and the proceeding reward. Alternatively, environments consisted of specific sequences of rewards including a pseudorandom sequence or an environment in which all rewards are of the same value. Different environments could be administered depending on the purpose of each experiment.

At the beginning of each task environment, participants were asked to register their mood. The task rounds then followed, where participants were asked to rate their mood when rewards were subtracted from or added to their total score. From now on, we will refer to these rewards as the task stimuli, with the sign of the stimuli being considered positive when a reward is added to, and negative when a reward is subtracted from, the participants' piggy bank. In each task round, the administration of each task stimulus lasted for approximately 3secs (Fig 1). When multiple environments were administered, the score was zeroed before the start of each environment.

As our task is new, we provide a table with the definitions of some key task terms (Table 1).

To register their mood, participants answered the question: "How happy are you at the moment?". by using a *mood-o-meter* which was similar to that used in previous experiments where momentary mood was studied [8, 9]. The mood-o-meter consisted of a vertical bar that did not have any mood values but contained signs and colors that helped participants better assess their mood. Specifically, smiling and sad faces were presented at each quarter of the meter to indicate the side where positive and negative mood should be registered, respectively. A color gradient from green to red was also used to indicate the mood range, with red used at the edges of the meter to signal extreme negative and positive mood values.

Participants could submit their mood by marking the height, on the mood-o-meter, that they believed better represented their momentary mood and the previously submitted mood

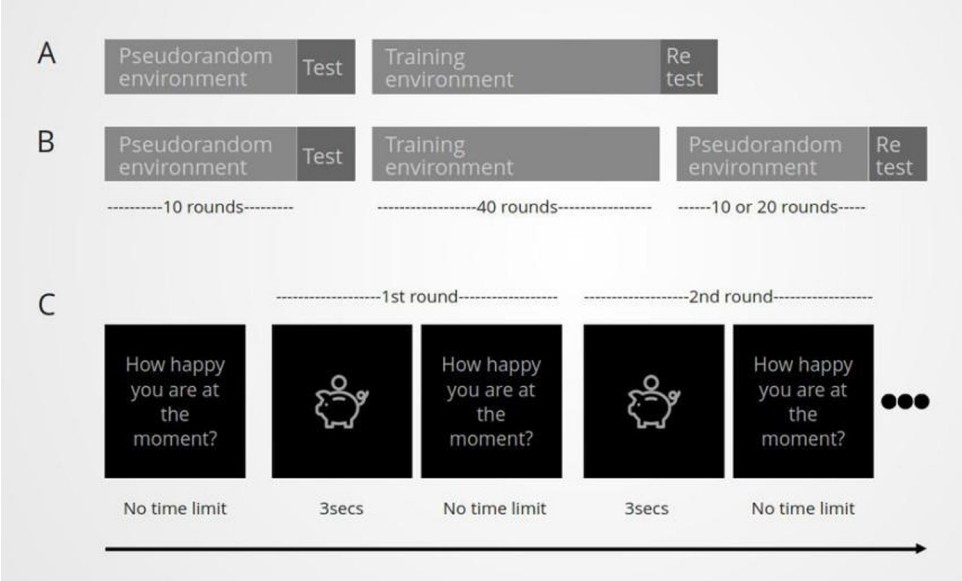

**Fig 1.** A graphic depiction of the experimental designs of Study 2(A) and Study 3(B) and the task (C) are shown in this Figure A. In Study 2, different task environments were examined to see whether mood could be trained. In this design a pseudorandom environment and the test sequence were followed by the training environment and the retest sequence. B. In Study 3, the duration of the training effect was also examined. To investigate that, the same experimental design was followed as in Study 2, but a pseudorandom environment of 10 or 20 task rounds was administered at the end of the training environment. C. In our task, participants are initially asked to rate their mood, and then the task rounds follow. Each round consists of the stimulus administration (3secs), followed by participants rating their mood.

was marked on the meter with the word "last" to help participants judge their momentary mood, in relation to their previously registered mood (for screenshots of the task, including the mood-o-meter see S1 File). No time restriction was imposed to participants concerning them rating their mood. This was done to avoid false ratings due to haste as well as to prevent causing stress and anxiety that can influence participants' mood ratings. Participants were instructed, however, prior to the beginning of the task, to complete the task with as few interruptions as possible.

**Table 1. Task terminology.**

| Task Terms | Definitions |
|---|---|
| Task Stimuli | They are points, in the form of coins, that are added (rewards) to or subtracted from (punishments) the participants' score |
| Task Rounds | They consist of a task stimulus administration, followed by the participants' mood rating in response to the stimulus |
| Task Environments | They are characterized by a specific relationship between mood and the subsequent task stimuli (ex. proportionate, paradoxical). In Study 2, we examined the effect that 16 task environments had on mood training. In Study 3, 2 task environments were selected to examine mood training and the duration of the training effect |
| Test/Retest Sequence | The test and retest sequences are identical and each one consists of up to three task rounds of relatively large punishments. Mood's slope in these rounds is calculated and used to estimate mood resilience to negative stimuli. The test and retest sequences are administered before and after the training environment, respectively |

A description of the main task and experimental design terms is included in this table.

Ethics approval for this study was obtained from the NIH Office of Human Subjects Research Protection.

## Participants

Participants for these studies were recruited online from the Amazon Mechanical Turk (MTurk) platform and the task was completed online. Data collection took place between May 2020 and August 2020. The Mturk Worker ID was used to distribute a fixed compensation for each participant who completed the task, for all three experiments. Participants were clearly instructed that the number of points they would win during the task would not be related to their final compensation which was fixed. This was done to prevent participants from exploiting the task to gain bigger compensation. A fixed amount deprived any motivation for participants to use the task to gather more points which would interfere with the design of our task and the quality of the data. The study population was adults of 18 years of age and older. Participants were not screened for eligibility, but only workers with an approval rate of 97% or greater and more than 1000 approved hits were allowed to participate to ensure data quality. Participants were selected to be located in the USA. Participants were excluded from performing the same experiment more than once and since multiple studies were conducted using the same task, participants were also excluded from any of our future studies. Prior to completing the task, online written informed consent was obtained from the participants and all data collected from MTurk were completely anonymized and unidentifiable. Ethical approval for these studies was received from the NIH Office of Human Subjects Research Protection (protocol number: P194594).

## First set of experiments: Validation of the task

### Study design of first set of experiments

The purpose of this set of experiments was to validate the task before using it as a tool to examine whether mood could become more resilient to negative stimuli. As the task was newly developed and had not been used before in another study, we wanted to test a) its efficiency in influencing mood, but also crucially b) the evidence we have that participants truly register their momentary mood, as opposed to providing ratings that are simply used to maximise rewards. For this purpose, in a first step, we created a task environment where rewards, at each task round, were administered in a pseudorandom order. In this environment, we intentionally placed three, consecutive task rounds (6th-8th) with large reward differences between the stimuli that would allow us to clearly observe mood's responsiveness to rewards and identify any unexpected patterns of responses. These task rounds were among the final ones to make sure that participants were already familiar with the task.

In a second step and in order to further validate that task and ensure that participants would not exploit it to gain maximum points, we conducted an experiment with a paradoxical task environment. Task exploitation was major concern in our experiment and is a threat to any experiment of this type, as in several of the task environments we used to examine mood resilience to negative stimuli in Study 2 and Study 3, there is a contingency between mood and rewards, as bad and/or good mood ratings are rewarded or punished. Before using these environments to examine whether mood could be trained to become resilient to negative stimuli, we wanted to ensure that participants would follow task instructions and register their subjective mood and not a mood rating that would increase their task score.

To do that, in this second step, we intentionally adopted a contingency between mood and rewards and created a paradoxical environment where good mood was proportionally punished and bad mood proportionally rewarded. As a result, good mood would yield a

punishment which would promote lower mood ratings, and negative mood would yield a reward and promote higher mood values. We expected that this would lead to an oscillatory mood course centered around zero mood. However, if participants were exploiting the task for profit, they would be continuously submitting low mood ratings as these, in this environment, are maximally rewarded. For analysis purposes, subjects were considered as task exploiters if they registered minimum mood ratings for three or more consecutive task rounds, as we considered it unlikely that a compliant subject would accidentally choose the minimum mood rating for that many rounds.

Finally, to examine whether participants understand the task well enough to exploit it, a modified version of the previous experiment was designed and the same task environment was administered. In this third step, we removed from the task's instructions any reference to mood. The mood-o-meter was replaced with a simple meter and participants were advised to play the task as they saw fit. We expected that in this task version, participants would submit extreme negative meter values as these were rewarded with maximum points.

## Analysis of first set of experiments

For all the experiments we plotted and observed the mood courses for each participant.

For the first step, we calculated the percentage of participants for whom mood was responsive to task stimuli. For this, we examined mood ratings for the three specific, consecutive rounds with large differences between the administered stimuli, as mentioned in our study design. This was a rather bold criterion that helped us reject participants whose mood was not responsive to stimuli as a responsive mood has a course that follows stimuli fluctuations.

For the second step, we identified task exploiters and measured mood oscillations. As stated in our study design, we defined as task exploiter anyone who registered minimum mood ratings for three or more consecutive rounds. Oscillations as a change in mood direction are generally expected to occur in all environments. For this paradoxical environment, we expected oscillations to have a specific characteristic. During each oscillation, we expected mood to go from positive values to negative values or vice versa. As the paradoxical environment consists of 40 task rounds, a maximum of 38 oscillations per participant was expected. Since any mood fluctuation in mood could produce oscillations, to validate this specific oscillatory pattern, we excluded task exploiters and compared the distribution of the number of produced oscillations in this task environment with the corresponding one in a task environment where stimuli were randomly administered to participants. The null hypothesis was that both the random environment and our paradoxical environment would not differ in the number of produced oscillations as we define them for this environment. In our analysis, we set no constraint concerning either the amplitude or the frequency of the oscillations.

To better validate our task's influence on mood ratings we applied, for each participant, a linear regression model. A linear regression model was selected as it does not assume any particular distribution for the effect of stimulus on mood. However, in order to ensure that taking into account both within and across person variances does not change the results, a mixed effects model was also tested (see S1 File). In these models, we tested, within each subject, whether the difference between each round's mood rating and the previous round's mood rating was associated with the presented task stimuli. The null hypothesis was that each difference in mood ratings would be independent of the task stimuli.

For the third step, where we expected participants to pursue maximum profit (see study design), the mean and median values that were registered from all participants were extracted to assess participants' ability to gain maximum rewards by submitting minimum meter values. A one-sided binomial test was performed on the observed frequency of exploiters to show that

the majority of participants exploit the task when they are not asked to provide mood ratings. The null hypothesis is that the rate of task exploiters would be 50% or less. To examine whether the participant's behavior differs based on whether or not participants are asked to register their momentary mood, we compared the results between the second and third experiment. We hypothesized that when participants were not asked to register their mood (third experiment), they would exploit the task for maximum gain. We conducted a chi-square test on the two-by-two contingency table of exploiters and non-exploiters from the two experiments. The null hypothesis was that the rate of task exploiters in the third experiment is equal to the rate of task exploiters in the second experiment.

## Results

The demographics for this set of experiments are presented in Table 2. For the first experiment, for the majority (80%) of our sample, the mood course was consistently responsive between participants, in the three tested rounds, such that changes in mood matched the signs of the presented stimuli (Fig 2A). In the second experiment, 10% of the participants were identified as task exploiters as per our criteria. For the remaining participants, our analysis showed that the number of expected oscillations (Fig 2B) was significantly different (Mann-Whitney U statistic = 3156.5, p < .01) in the paradoxical task environment (N = 164, mean = 14.31, median = 12, IQR = 23.25) compared to the random one (N = 29, mean = 5.89, median = 6, IQR = 7). The results of the linear regression model showed that, after correction for false positives, for 91% (CI = 0.86–1.00) of participants' mood was significantly influenced by the task stimuli.

These findings are in line with our initial hypothesis and taken together with the previous finding, they indicate that the task is efficient in influencing mood. Participants avoid random ratings when registering their momentary mood and in their vast majority (90%, p<0.000001, one-sided binomial on non-exploiters frequency under the null hypothesis of 50% exploiter frequency) did not exploit the task for maximum gain. In the third experiment, where participants were not specifically instructed to register their momentary mood, the majority (93%, p < .0001) submitted extreme, low meter values (mean = -0.8, median = -0.98) that yield maximum points for this task (Fig 2B). A statistically significant difference (Chi-square statistic = 95.89, p<0.00001) was observed when the rate of exploiters in this experiment was compared to that of the second experiment.

In conclusion, we have shown that:1) For most participants, mood ratings are responsive to the presented stimuli. 2) Most participants do not exploit the task for maximal gain when asked to provide their mood rating. 3) Participants do understand how to exploit the task and can do so when the task instructions do not ask them to provide a mood rating.

## Second set of experiments: Mood training and identification of efficient experimental environments for training

These experiments were conducted to test whether experimental environments could train mood to promote mood resilience to negative stimuli, meaning that mood, following a successful training, would be less susceptible to administered negative stimuli.

## Study design of second set of experiments

Our task design included an environment of a pseudorandom sequence (10 rounds) followed by the test sequence and 40 rounds of the training environment followed by the retest sequence (Figs 1 and 3). The test and retest sequences were identical and consisted of three consecutive task rounds with punishments of values (-40, -45, -50) near or at the upper limit of punishment

**Table 2. Demographic data.**

| | Environment | N | Female(%) | Age (min-max, Mean, StD) |
|---|---|---|---|---|
| Study 1 | pseudorandom | 187 | 56 | 19–62, 41.0, 8.86 |
| | paradoxical | 182 | 53 | 18–64, 37.6, 10.9 |
| | maximum gain | 26 | 46 | 20–58, 39.8, 9.6 |
| Study 2 | proportional reward of good mood | 39 | 44 | 18–53, 36.3, 9.2 |
| | reverse proportional reward of good mood | 37 | 43 | 19–54, 38.3, 8.3 |
| | constant medium reward of good mood | 31 | 55 | 19–51, 35.9, 8.3 |
| | constant maximum reward of good mood | 28 | 43 | 18–61, 42.1, 12.3 |
| | proportional punishment of good mood | 48 | 58 | 18–57, 36.6, 12.3 |
| | reverse proportional punishment of good mood | 33 | 52 | 18–58, 35.4, 10.0 |
| | constant medium punishment of good mood | 28 | 43 | 19–57, 36.2, 8.8 |
| | maximum punishment of good mood | 25 | 40 | 20–59, 40.6, 10.1 |
| | proportional punishment of bad mood | 38 | 42 | 20–52, 38.4, 8.1 |
| | reverse proportional punishment of bad mood | 44 | 48 | 18–59, 35.7, 10.9 |
| | constant medium punishment of bad mood | 30 | 40 | 18–61, 37.3, 9.7 |
| | maximum punishment of bad mood | 26 | 58 | 23–52, 37.1, 8.0 |
| | proportional reward of bad mod | 29 | 62 | 18–56, 36.7, 9.0 |
| | reverse proportional reward of bad mod | 31 | 58 | 20–58, 38.3, 9.4 |
| | constant medium reward of bad mood | 32 | 44 | 18–55, 38.0, 8.6 |
| | maximum reward of bad mood | 24 | 54 | 20–53, 35.0, 8.7 |
| Study 3 | reverse proportional reward of good mood, 10 rounds | 30 | 40 | 26–53, 40.4, 7.4 |
| | maximum reward of good mood, 10 rounds | 36 | 47 | 19–59, 40.2, 9.9 |
| | reverse proportional reward of good mood, 20 rounds | 54 | 54 | 20–59, 39.5, 9.1 |
| | maximum reward of good mood, 20 rounds | 59 | 49 | 18–63, 38.1, 10.7 |

The demographic data for the three studies are included in this table. The total number of participants, the gender balance and mean age are presented for each experiment/environment that was examined for the three studies. StD: Standard deviation

values used throughout the rest of the task, in which the range of stimuli values was -50 to +50 (Table 2). The choice of up to three task rounds for the test and retest sequence was not based on previous literature as this is a new task, our rationale for this choice however was based on the following. Both the test and retest sequences consist of negative stimuli and as a result they needed to be relatively short in length to avoid extreme loss of points that would, in itself, impact mood. Additionally, as with all training effects, we expected that our task effect would also wane with time and thus a relatively long sequence would not be representative of the training effect that our training environments would have on mood.

For this experimental design, two possible scenarios need to be accounted for and controlled: A) Before completion of the test or retest sequence, mood values could already be so low that for the proceeding big, negative stimuli, mood's expected fall would be restricted by the mood-o meter range. This would lead to floor effects. To avoid this scenario, if the registered mood values fall below the lower quarter of the mood-o-meter, after the first or second stimulus of the test or retest sequence, the sequence is stopped. B) Before delivering the test or retest sequence mood could already be in the negative half of the mood-o-meter. This would reduce the available range for mood to fall. To avoid this, if prior to the administration of the test or retest sequence mood is negative, a maximum of two rewards proportionate to its value are administered. The purpose of this is to push mood to the positive half of the meter.

The design of the control experiment is identical to that described above, but for the training environment randomly generated rewards were used.

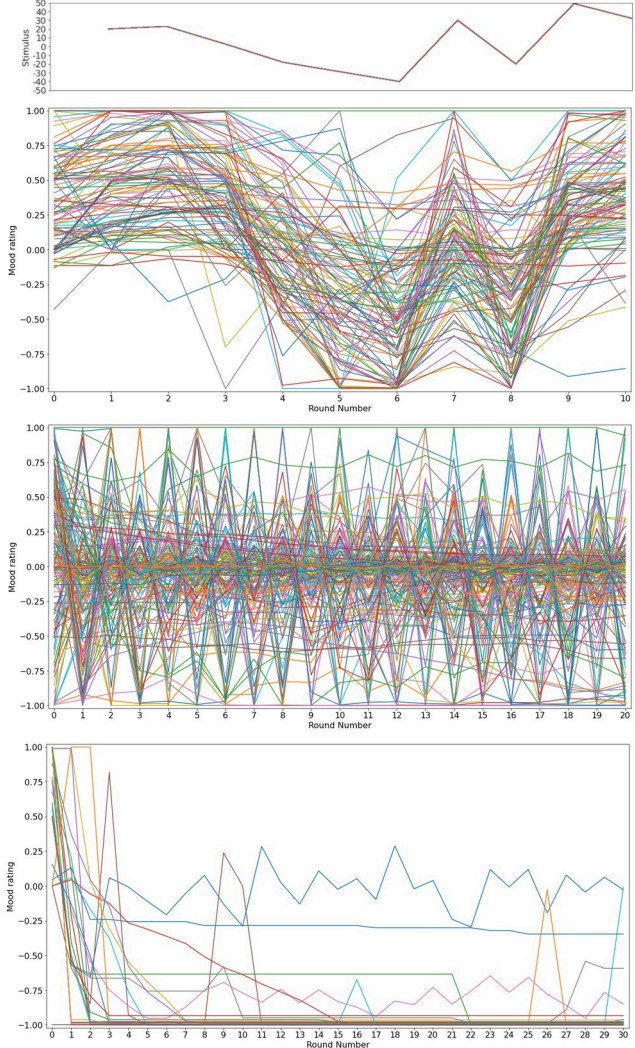

**Fig 2. Task validation.** A. In the pseudorandom environment similar mood courses are observed for the majority of participants. The change in mood's direction appears to follow the task stimuli which are depicted in the graph above. B. In the paradoxical environment, we observe an oscillatory pattern of mood responses indicating that participants are not exploiting the task for maximum gain. C. When any mention of mood is removed from the task, maximum gain environment (bottom graph), the majority of participants (93%) register extreme negative values which yield maximum task points.

## Training environments of second set of experiments

Each training environment consists of 40 rounds. As the reward is a function of mood in these environments, we visualize the relationship between reward and mood graphically: mood is placed on the x axis and reward is placed on the y axis (Fig 4). In this two-dimensional space, two quadrants belong to positive and two quadrants to negative mood. For each quadrant, we distinguish four possible training environments characterized by different linear relationships between mood and reward. These relationships include A. a proportional one where mood and rewards have an increasing linear relationship; B. a reverse-proportional one where mood magnitude and rewards have a decreasing linear relationship; C. one where a medium reward is always administered and D. one where a maximum reward is always administered. We examined each of these four relationships in each quadrant of the space for a total of 16

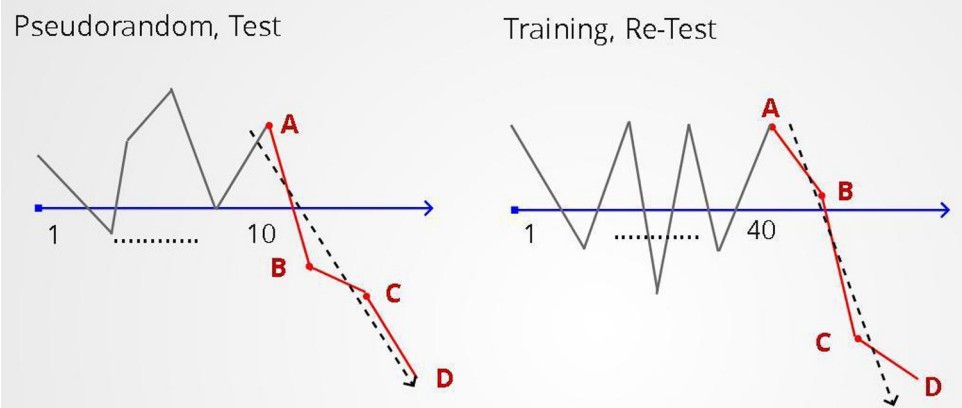

**Fig 3. Mood training.** Our task design included a pseudorandom task environment (10 rounds) followed by the test sequence and 40 rounds of the training environment followed by the retest sequence. The test and retest sequences are identical and consist of three consecutive, negative stimuli of large value. In this Figure A: represents the mood measurement before the administration of the test/retest stimuli sequence. B: represents the mood measurement after the administration of the first stimulus of the test/retest sequence. C: represents the mood measurement after the administration of the second stimulus of the test/retest sequence. D: represents the mood measurement after the administration of the third stimulus of the test/retest sequence.

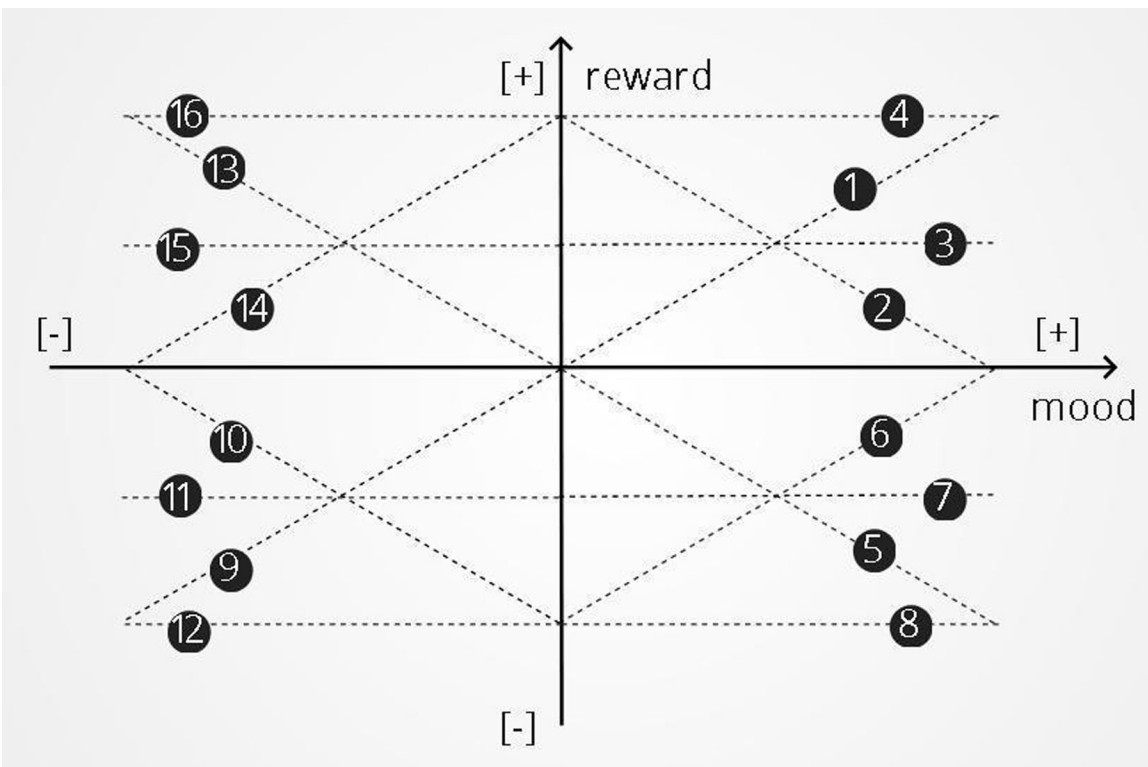

**Fig 4. Mood and reward.** We represent mood and reward on a two-dimensional space. In this space mood is placed on the x axis and reward is placed on the y axis. Two quarters of this space belong to positive and two quarters to negative mood. For each quarter of this space we distinguish four possible training environments characterized by different mood and reward linear relationships. These relationship include: A proportional one where mood and rewards have an linear relationship—lines 1, 5, 9, 13. A reverse-proportional one where mood and rewards have a negative linear relationship- lines 2, 6, 10, 14. An environment where a medium reward is always administered—lines 4, 8, 12, 16. An environment where a maximum reward is always administered—lines 3, 7, 11, 15.

training environments, with each environment being characterized by a combination of quadrant and type of linear relationship. In each of these training environments, if the participant reported mood outside of the region being tested, random rewards or punishments were administered to push mood toward the appropriate region. This occurred more commonly when punishing good mood or rewarding bad mood. For example, when rewarding bad mood (the top left quadrant of Fig 4), would lead to positive mood ratings. Since the goal of the experiment is to train negative mood, random punishments would be delivered to push mood back to negative values.

### Filters for participant selection for the second set of experiments

Two filters were applied to ensure that participants were performing the task according to our instructions and were registering their true momentary mood as well as to ensure data quality, which is often compromised when data are collected online (for example: [13–16]). Participants were allowed to proceed with the experiment only if during the start of the pseudorandom task environment they were responsive to mood fluctuations at three specific, consecutive rounds; B) Participants were not allowed to proceed with the task if during the test sequence the last registered mood was higher than the mood before the beginning of test sequence. The same rule applied to the retest sequence. Participants that did not pass these filters, had a greater possibility of either not playing the task right or their mood being unresponsive to change or responding paradoxically and thus deviated from common forms that we cannot at this point test with our experiment. Based on these filters, the rejection rates were approximately 30%-40% (see Table 1 in S1 File). We acknowledge that this may indicate that our filters are very strict, but due to the exploratory nature of our experiments, we did not want to jeopardize the quality of our data.

### Analysis of second set of experiments

We next computed mood's decline (downward slope) for the test and the retest sequences, which allowed us to determine whether our experimental environments had a training effect on mood so as to make it more resilient to negative stimuli, for each participant and for each of the 16 training environments. The slope was computed based on all mood ratings of the test and retest sequence. We then conduced a simple linear regression with the slope of the retest sequence as the dependent variable. The independent variables in our analysis were the slope of the test sequence and the registered mood value before the start of the retest sequence. We also included a dummy independent variable to indicate whether the experiment included an actual training environment or was a control experiment.

$$retest\_slope \sim test\_slope + mood\_before\_retest + is\_training\_group$$

Statistical significance was assessed by examining the beta coefficient of this dummy variable as this represents the added contribution of the specific training environment, in contrast to the control environment. Since these experiments were exploratory, multiple comparisons correction was not applied.

### Results

Six task environments were identified where there was a significant effect of training on mood, compared to the control experiment. In five of these environments (represented by green lines in Fig 5) there was a statistically significant positive effect of training on mood resilience to negative stimuli while in one task environment there was a statistically significant negative

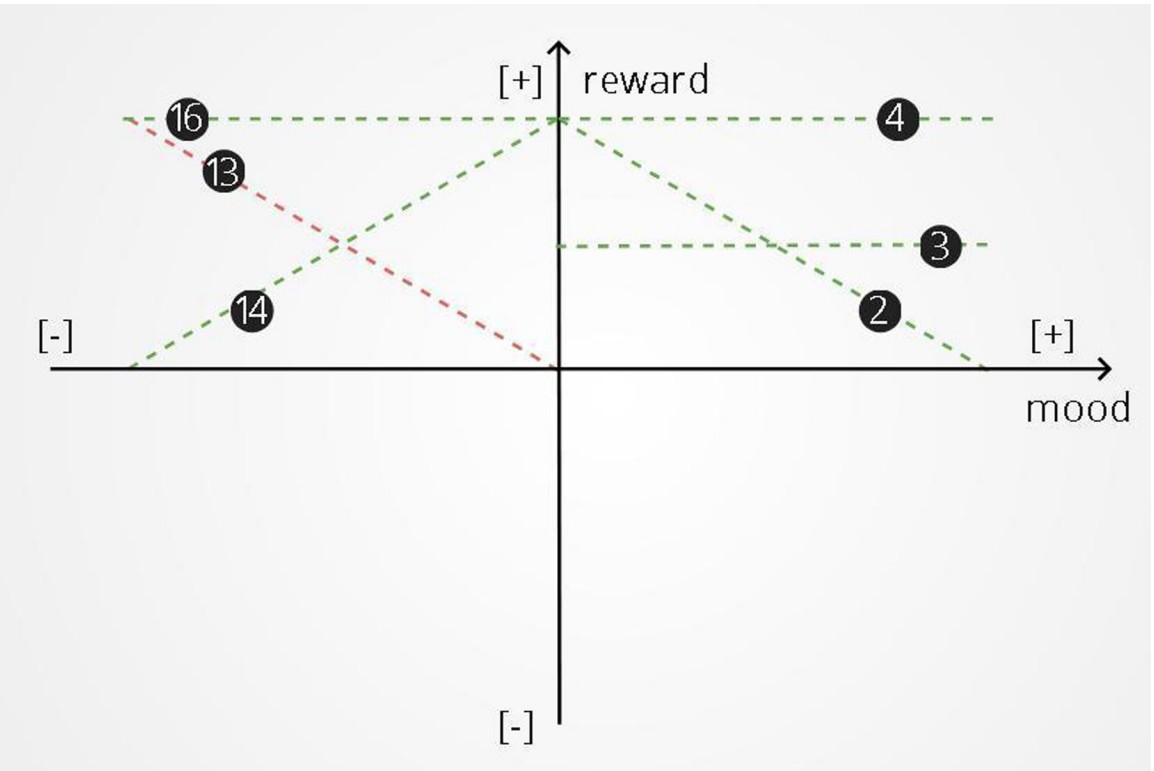

**Fig 5. Successful environments for mood training: Two environments where good mood receives the maximum and medium reward—lines 4 and 3.** A reverse-proportional environment—line 2. An environment where bad mood always receives the maximum reward—line 16. A reverse-proportional environment—line 14. A proportional environment—line 13.

effect of training on mood resilience to negative stimuli. Specifically, a statistically significant reduction of the mood slope was observed in A) the environment where any good mood value receives the maximum reward ($\beta = 0.465$, SE = 0.125, $p < .001$); B) the task environment where any good mood values receive the medium reward ($\beta = 0.263$, SE = 0.129, $p < .05$); C) the reverse-proportional environment where good mood and reward have a negative linear relationship ($\beta = 0.411$, SE = 0.105, $p < .001$); D) the task environment where bad mood always receives that maximum reward ($\beta = 0.201$, SE = 0.079, $p < .05$) and E) the reverse-proportional task environment where bad mood and rewards have a negative linear relationship ($\beta = 0.235$, SE = 0.068, $p < .01$). A statistically significant increase of the mood slope ($\beta = 0.17$, SE = 0.074, $p < .05$) was observed in the environment where bad mood and rewards have a proportional relationship (represented with a red line in Fig 5). All the training environments that were examined for this study, the demographic data for each environment and the results of mood training are presented in Tables 2 and 3.

## Third set of experiments: Testing the resilience of the training effect after multiple number of rounds

To assess whether the training effect would last over time and replicate our previous findings, we isolated the two most effective training environments from the previous set of experiments and examined the resilience of the effect after 10 and 20 rounds, corresponding to approximately 1 and 2 minutes, respectively.

**Table 3. Results of the second set of experiments.**

| Environment | β-coef | CI(2.5,97.5) | SE | p value |
|---|---|---|---|---|
| proportional reward of good mood | 0.194 | -0.02, 0.408 | 0.107 | 0.074 |
| **reverse-proportional reward of good mood** | **0.411** | **0.201, 0.621** | **0.105** | **<0.001** |
| **medium reward of good mood** | **0.263** | **0.006, 0.521** | **0.129** | **0.045** |
| **maximum reward of good mood** | **0.465** | **0.216, 0.715** | **0.125** | **<0.001** |
| proportional punishment of good mood | -0.038 | -0.173, 0.098 | 0.068 | 0.582 |
| reverse-proportional punishment of good mood | -0.021 | -0.175, 0.132 | 0.077 | 0.783 |
| medium punishment of good mood | -0.059 | -0.216, 0.097 | 0.078 | 0.453 |
| maximum punishment of good mood | 0.044 | -0.108, 0.076 | 0.075 | 0.564 |
| proportional punishment of bad mood | 0.146 | -0.024, 0.317 | 0.085 | 0.091 |
| reverse-proportional punishment of bad mood | -0.06 | -0.206, 0.085 | 0.073 | 0.414 |
| medium punishment of bad mood | 0.137 | -0.051, 0.325 | 0.094 | 0.151 |
| maximum punishment of bad mood | 0.128 | -0.061, 0.316 | 0.094 | 0.181 |
| proportional reward of bad mood | -0.17 | -0.319, -0.021 | 0.074 | 0.025 |
| **reverse-proportional reward of bad mood** | **0.235** | **0.099, 0.370** | **0.068** | **<0.001** |
| medium reward of bad mood | 0.013 | -0.14, 0.166 | 0.076 | 0.869 |
| **maximum reward of bad mood** | **0.201** | **0.041, 0.361** | **0.079** | **0.014** |

In this exploratory set, all 16 environments were tested for their efficacy in training mood and the results are presented in here. For each environment and for each participant we computed, mood's falling slope for the test and the retest sequences. We then conduced a simple linear regression including as factors to our analysis, the slope of the test sequence and the registered mood value before the start of the retest sequence. We also accounted for whether the experiment included an actual training environment or was a control experiment (N = 41, Females 54%, Mean Age = 42±10).

## Study design

The experimental design we followed was identical to the one described in the second set of experiments including the same filters for participation The only difference was that the training environment was followed by a pseudorandom task environment (Figs 1 and 6). The number of rounds of this pseudorandom environment depended on the duration of the training

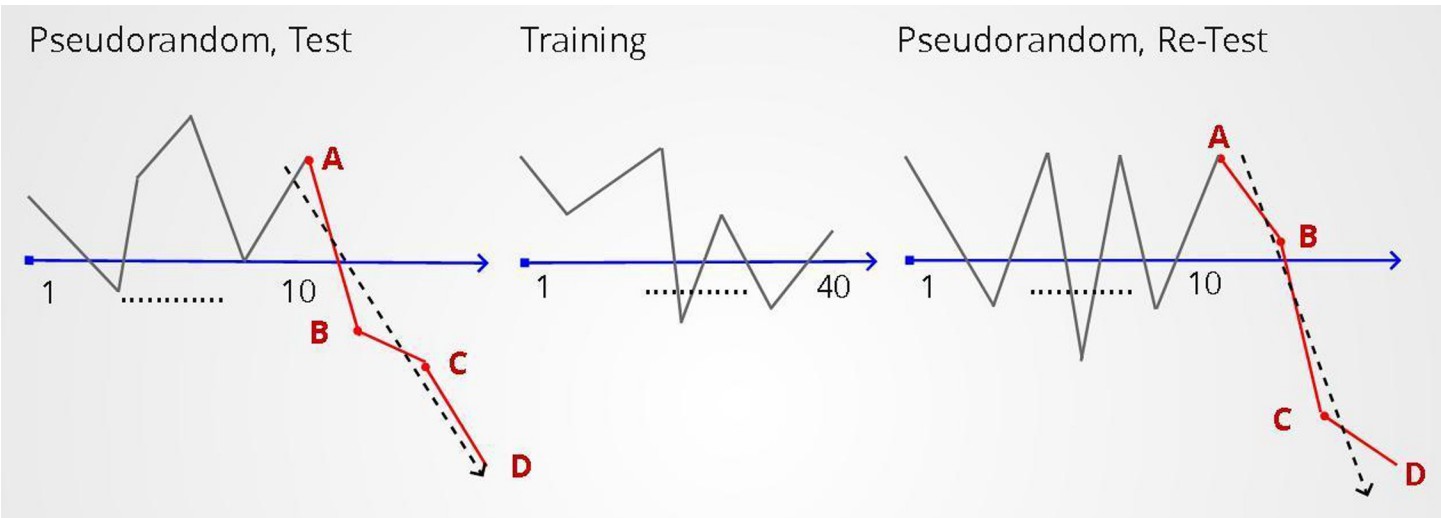

**Fig 6. Mood training resilience.** To test the resilience of the mood training over multiple task rounds, we introduce a new environment of pseudorandom sequences after the training environment. In this study, the pseudorandom environment consisted of 10 and 20 rounds.

effect that we wish to test (10 or 20 rounds). This pseudorandom was then followed by the retest sequence. The training environments used for these experiments were A) the environment where good mood is associated with the maximum positive reward and B) the reverse-proportional environment where positive rewards have a linearly decreasing relationship with good mood. These two environments were identified in the previous study as the ones with the greatest training efficiency. We checked the resilience of our training effect after 10 and 20 rounds, compared to the corresponding control experiments.

## Analysis

The same analysis was followed as in the previous study. For these analyses, Bonferroni correction for multiple comparisons was applied.

## Results

A statistically significant reduction of the mood slope, compared to the control environment was detectable 10 and 20 task rounds after the training environment when the environment where good mood is associated with the maximum positive reward (10 rounds: β = 0.228, SE = 0.080, p < .01, 20 rounds: β = 0.307, SE = 0.068, p < .001) that was selected from the previous set of experiments was examined (Tables 4 and 5). After Bonferroni correction, no statistically significant effects were detected for the reverse-proportional environment. The demographic data for these experiments are presented in Table 2.

## Discussion

In this study we aimed to train mood to build mood resilience to negative stimuli. For that purpose, we developed a new experimental design which allowed us to examine various training environments.

Given the exploratory nature of our study and the novelty of our task, in our initial set of experiments we tested and confirmed our task's efficiency in influencing mood. Previous experiments have successfully employed similar methods to study mood and identify the mechanisms that influence mood dynamics [8, 9]. Our aim, however, was to develop a tool to examine the potential of mood to be trained and as such we wanted to be sure that mood in our task would be responsive to stimuli and participants would follow task instructions and not exploit the task. Additionally, and in contrast to previous tasks that examine mood in relation to rewards and punishments [8, 9, 17, 18], in our paradigm participants are not required to perform any action to receive or lose points. Despite our deviations from previous task designs, we showed that in our task, mood follows the stimuli fluctuations while the majority of participants do not exploit the task for maximum gains. This is important as we aimed for the task to be able to capture the participants' mood and thus allows us to explore mood training and avoid participants registering random mood values.

The second set of experiments was done to examine our main experimental question of whether the environment could be used to develop mood resilience to negative stimuli. In that set of experiments, we tested sixteen different experimental task environments. In five of these environments, mood training promoted resilience to negative stimuli, while one environment promoted mood susceptibility to negative stimuli. Deviating from our initial hypothesis, which included a role for punishment in mood training, we found that only environments where mood is rewarded have statistically significant effects in training mood; however, it remains a possibility that a higher-powered replication could discover additional environments based on punishments that are also efficient in training mood.

**Table 4. Mood resilience to negative stimuli after 10 rounds.**

| Environment | β-coef | CI(2.5,97.5) | SE | p value (adjusted) |
|---|---|---|---|---|
| reverse-proportional reward of good mood | 0.175 | 0.007, 0.343 | 0.083 | 0.041 (0.164) |
| **maximum reward of good mood** | **0.228** | **0.067, 0.388** | **0.080** | **0.006 (0.024)** |

The resilience of mood training was further measured in a novel sample for the two environments that had the greatest training efficacy as per Study 2. The effect of each environment after 10 rounds of a pseudorandom sequence is shown. Both environments were compared to a similar control experiment (N = 33, Females 42%, Mean Age = 40±9).

One reason that the environments utilizing rewards for training might have stronger effects (and thus have higher power to be detected in our limited-sample size study) is that rewards might be more effective than punishments in learning tasks. Several research studies have shown that in learning tasks, especially those involving procedural learning, and under specific experimental settings, rewards and punishments equally improve task performance but punishments, unlike rewards, fail to enhance learning (for example studies see [19–22]). This could be attributed to rewards activating neuronal mechanisms that are more beneficial to learning and subsequent memory consolidation. Alternatively, punishments have been shown to reduce the quality and quantity of information that is retained, at least in decision-making tasks [22, 23].

Another observation from our second set of experiments was that all environments that were successful in training mood were these where mood values close to zero are equally or better rewarded compared to more extreme values. In biology and neuroscience, mood has long been viewed as a mechanism to promote survival and help individuals better adapt to their environment [5, 24, 25]. As such, mood states should be cost and energy effective but also amenable to change. It is thus possible that when mood is rewarded, good and bad values that are not extreme but closely above and below zero, are evolutionary preferred as these would offer the optimal balance between rewards and cost but would also allow for a bidirectional change and adaptability to the environment. This observation is in line with recent findings from large studies showing that mood could be subject to homeostatic mechanisms probing individuals to behaviors that would help them retain their normal mood state [26].

Overall, in this second set of experiments we showed that the environment, within the context of our task, could be harnessed to influence mood's future responses to stimuli, as we showed that five of the sixteen environments promoted mood resilience to negative stimuli, while in one task environment, mood sensitivity to negative stimuli was observed. The fact remains however, that the majority of the task environments that we examined, failed to statistically alter mood's responses to future negative stimuli. As the purpose of our study was to examine whether mood could be trained, we did not have such an experimental design that would allow us to examine the factors that influence mood training. We cannot, as a result,

**Table 5. Mood resilience to negative stimuli after 20 rounds.**

| Environment | β-coef | CI(2.5,97.5) | SE | p value (adjusted) |
|---|---|---|---|---|
| reverse-proportional reward of good mood | 0.118 | -0.024, 0.259 | 0.071 | 0.102 (0.408) |
| **maximum reward of good mood** | **0.307** | **0.172, 0.441** | **0.068** | **<0.001 (<0.001)** |

The resilience of mood training was measured in two environments that had the greatest training efficacy as per Study 2. In one of the environments the effect was detectable after 20 rounds of the pseudorandom sequence. Both environments were compared to a similar control experiment (N = 62, Females 44%, Mean Age = 34 ±10).

offer an explanation as to which characteristics of these ten task environments influenced mood training and we discuss this further in our limitations.

In order to test the replicability of our findings on mood training and examine whether mood resilience to negative stimuli would be detectable after 10 and 20 rounds (1 and 2 minutes) of a pseudorandom environment, we conducted a third and final set of experiments. For these experiments we used the two environments where training had the greater effect on mood resilience to negative stimuli and showed that mood resilience is present at least up to 2 minutes (20 rounds) after training. A recent experiment on the influence of past and previous events on mood, had showed that past events still carry a lot of weight when momentary mood is assessed [8]. In accordance with these findings, we expected that mood training which occurred in an initial training environments could influence momentary mood even after several rounds.

To strengthen our findings and expand our knowledge on mood, several future experiments and analyses should be conducted. The task and our data on mood training would profit from a more detailed analysis of the task parameters including for example, the optimal number of rounds for training and different reward scales between the testing and training environments.

In tandem with further experiments exploring different possible environments for mood training, a computational modelling approach would allow us to characterize the parameters that are important for mood training. These parameters could then be used to construct a novel environment with maximum effects on mood. The generalizability of our task should also be tested by using other types of stimuli, such as pictures with positive and negative valence. Since different types of stimuli can activate different neuronal pathways during reward-based learning, it is important to know whether the observed effects on mood are specific to our stimuli or can also be applied to other types of rewards. Finally, the task could be used to examine the effect of positive stimuli on mood. This could be particularly beneficial for clinical populations and more specifically patients with mood disorders.

Our study has several limitations. Data collection for the studies involving our task was conducted online using MTurk. The use of online platforms allows for fast data collection and large experimental samples. However, several checks need to be performed to ensure data integrity as highlighted by other studies (for example see [13–16]). We applied two filters to guarantee data integrity and make sure that our participants were performing the task based on our instructions and no random choices were made. However, we cannot be certain that we managed to avoid all the risks that are linked to online data collection. Additionally, since this is an online sample, rather than a random sample from the general population, we caution on inferences about the generalizability of our findings. Concerning our task design, the word "last" was shown on the mood-o-meter to help participants better assess their momentary mood, in relation to the previously inputted mood value. It is possible, however, that this indication could have further enhanced or somehow impacted expectancy effects, influencing participants into mood rantings strongly based on the previous stimulus. Moreover, since our study was exploratory, we investigated mood training using the simplest experimental environments. Out of the sixteen environments we tested, six were significant in training mood. However, we cannot exclude that other environments could also have a significant effect on training mood if a bigger sample was provided. Environments that are characterized by more complex relationships between mood and rewards may have even greater efficacy in training mood. For some of the mood-reward relationships we targeted for mood training, we needed to guide the participant's mood to the desired mood valence. As a result, in these environments some rounds were spent to adjust mood, which reduced the number of rounds that were attributed to mood training.

Moreover, we cannot offer any insight on the specific mechanisms that underlie mood training and consequently the duration of its effect on mood over time. For example, we cannot exclude the influence that the total score per se could have on mood training. Finally, as the participants' score was zeroed when participants were moving from one task environment to the next, in the second and third set of experiments, we cannot exclude that this influenced participants' mood and perhaps mood training. As this process, however, was applied to all participants in all experiments, including the controls, the effect that this might have, would be accounted for in our analysis. In the future, experiments need to be performed with and without this manipulation, to examine the effect that zeroing the score might have on task performance. Finally, it is standard practice, for psychiatric assessments as well as the majority of studies that examine mood, to only rely on subjective mood ratings. Future studies could complement subjective mood measures with other assessments, including physiological metrics (heart rate, pupillometry), that could help further characterize mood and mood resilience to negative stimuli.

Despite the several limitations, our study results show that our task could be used as an effective tool to study momentary mood and propose a way to investigate mood resilience to negative stimuli. Future studies are needed to further fortify our findings and the usefulness of our methods.

## Supporting information

**S1 File.**
(DOCX)

## Author Contributions

**Conceptualization:** Vasileios Mantas.

**Data curation:** Vasileios Mantas, Dylan M. Nielson.

**Formal analysis:** Vasileios Mantas, Vasileia Kotoula, Charles Zheng.

**Methodology:** Vasileios Mantas, Dylan M. Nielson, Argyris Stringaris.

**Supervision:** Argyris Stringaris.

**Writing – original draft:** Vasileios Mantas, Vasileia Kotoula.

**Writing – review & editing:** Charles Zheng, Dylan M. Nielson, Argyris Stringaris.

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
