## [Decision Letter · Decision Letter 0]

20 Jun 2023

PONE-D-23-07979An experimental approach to training mood for resiliencePLOS ONE

Dear Dr. Mantas, dear Dr. Kotoula, 

Thank you for submitting your manuscript to PLOS ONE. After careful consideration, we feel that it has merit but does not fully meet PLOS ONE’s publication criteria as it currently stands. Therefore, we invite you to submit a revised version of the manuscript that addresses the points raised during the review process.

Please consider especially the following recommendations: Include the sample size calculationavoid redundancy in the introductionproof-read the manuscript regarding sentence-structure and punctuation"mood resilience" might be not the correct term, but rather being resilient to an appearing affective state or a habituation process? You also describe it as resisting "the effects of exposure to adverse stimuli", but not mood itself? Please clarify.Please submit your revised manuscript by Aug 04 2023 11:59PM. If you will need more time than this to complete your revisions, please reply to this message or contact the journal office at plosone@plos.org. Please include the following items when submitting your revised manuscript:A rebuttal letter that responds to each point raised by the academic editor and reviewer(s). You should upload this letter as a separate file labeled 'Response to Reviewers'.A marked-up copy of your manuscript that highlights changes made to the original version. You should upload this as a separate file labeled 'Revised Manuscript with Track Changes'.An unmarked version of your revised paper without tracked changes. You should upload this as a separate file labeled 'Manuscript'.

We look forward to receiving your revised manuscript.

Kind regards,

Johanna Löchner

Academic Editor

PLOS ONE

Journal Requirements:

This work was funded by the Intramural Research Program of the National Institute of Mental Health, part of the National Institute of Health. The funder had no role in the design and conduct of the study; collection, management, analysis and interpretation of the data; preparation, review and approval of the manuscript; or decision to submit the manuscript for publication. 

The views expressed in this article do not necessarily represent the views of the National Institute of Health, the Department of Health and Human Services or the United States Government. 

Additional Editor Comments:

Dear Dr. Mantas, dear Dr. Kotoula,

thank you for submitting your manuscript "An experimental approach to training mood for resilience" to PLOS one.

We finally received two reviews with minor and major revision recommendations. Please read and respond carefully to the reviewer's comments, before you submit a revised version.

Kind regards,

Johanna Löchner

Reviewers' comments:

Reviewer's Responses to Questions

**Comments to the Author**

1. Is the manuscript technically sound, and do the data support the conclusions?

Reviewer #1: Yes

Reviewer #2: Partly

2. Has the statistical analysis been performed appropriately and rigorously? 

Reviewer #1: Yes

Reviewer #2: Yes

3. Have the authors made all data underlying the findings in their manuscript fully available?

Reviewer #1: Yes

Reviewer #2: Yes

4. Is the manuscript presented in an intelligible fashion and written in standard English?

Reviewer #1: Yes

Reviewer #2: Yes

5. Review Comments to the Author

Reviewer #1: The goal of this study was to test a method for training mood resilience, namely a way of decreasing mood sensitivity to negative stimuli such as losses in a gambling task. To this end, a series of experiments were conducted to first generate and then confirm hypotheses regarding the respective impacts of different task environments. The authors argue that this type of basic research may offer a future avenue for training mood and enhancing well-being.

They present a methodologically sound experimental paper and a well-written manuscript. The number of sub-experiments and progression from exploratory to confirmatory point towards a rigorous research process.

I have only minor suggestions to polish an already good manuscript.

Introduction:

- Good review of previous studies in the field and clear mention of what the current study adds. The research question is clearly stated and the reasoning that leads up to the hypotheses seems sound.

Method:

- Overall, the methods section could benefit from another proof-reading (including punctuation and sentence structure), as it contains a few sentences which are difficult to understand. However, I appreciate the clear statements on what was expected for each experiment/analysis, as it makes following the experiments easier.

- On page 5, it is stated “administration of task stimulus lasted approximately 3 seconds”: What does “approximately” mean in this context, did this time window differ between experiments/participants/…?

- Marker for “last” on the mood-o-meter (p. 7): The manuscript states this was added to help participants focus on their momentary mood, compared to their previously registered mood. Could this not also induce expectancy effects, e.g. if participants assume they should feel better after a reward, but wouldn’t actually indicate a better mood without the marker of their previous response?

- Participants: Was there a maximum age allowed to enter? Are any other demographic data available for the sample? Did the authors conduct a power analysis, and if not, what is the reasoning and how else did they arrive at this sample size (i.e., how was determined when to stop data collection)? How was the distribution between studies decided, and more importantly, how was the distribution between task environments within a study decided (were participants randomized or what determined their allocation)?

Results:

- Both the analysis and results sections seem very detailed to me and contain clear explanations as to why certain steps were or were not taken (e.g., why exploratory analyses were not corrected for multiple testing whereas confirmatory analyses were indeed corrected).

- It is surprising to me that such high percentages (80% and 91% in study 1) of participants showed mood responsiveness to a relatively simple reward/punishment task (although this might be because it is not my field of expertise). If I understood it correctly, the criterion for classifying participants as non-responsive was if they gave the same rating three times in a row. How much did ratings fluctuate (e.g., range of ratings)? And do the authors have any hypotheses for why some individuals might be more responsive to their task than others?

Discussion:

- The authors offer a critical discussion of their results, including study limitations, without diving into interpretations that would not be suitable to the experimental set-ups. Similar to the introduction, the discussion section benefits from clear reasoning and is easy to follow. I also appreciate that implications are discussed mostly in relation to future research, seeing as clinical implications of such basic research would not be appropriate.

- Even though the manuscript already contains discussion of several limitations, I would recommend adding an additional 1-2 sentences on generalizability of the results considering this particular sample.

Reviewer #2: The aim of the present exploratory study was to examine whether mood can be trained to become more resilient to future negative events. The authors developed a new series of experiments, in which participants rated their current mood state when rewards or punishments were administered. In the first series of experiments, the task was validated. In the second and third series of experiments, the authors examined which different task environments are efficient in training mood to become more resilient to negative stimuli.

The approach taken is innovative and the results provide an interesting starting point for future research that aims to examine the effectiveness of mood training in healthy and clinical populations including individuals with depression. Below are a number of minor and major points that should be addressed prior to publication:

Abstract:

Please be more concise regarding the methods and results of the current study. For example, please clearly describe which environments were found to be effective. Please e.g. also clearly state that this was an online study conducted in healthy volunteers (including information on the number of participants).

Introduction:

The introduction should be shortened. For example, there is some redundancy in the description of the different series of experiments on page 4.

Methods:

The question: !How happy are you at the moment?” is not neutral. Why did the authors not use a more neutral question regarding current mood?

Participants:

Very little sociodemographic information is provided (e.g. SES). Therefore, the generalizability of the findings remains unclear. This should be stated as a limitation. What was the age range of participants (18 to ?).

How exactly did the authors ensure that participants only participated once? Was the IP address tracked?

The fact that participants were instructed that the number of points during the task was unrelated to the final compensation limits the ecological validity of this experimental work. The authors should discuss how this aspect might have influenced behavioral responses.

The authors detail that task exploiters were excluded. Did this group differ from the remaining sample in terms of sex, age, etc.? One might argue that “task exploiters” were individuals who understood the contingency between their ratings and outcome and responded accordingly.

It is unclear how the necessary sample size was determined. Did the authors calculate a priory power calculation? This seems particularly important as it was found that only environments were mood was rewarded showed significant effects in training mood.

As the authors relied on a self-report measure of mood, it would have been beneficial to collect indices of social desirability and control for this aspect.

Table 1: Please report means and SDs with one decimal place.

Results: Please report <.001 instead of e.g. p<.00001 or p<000019.

Discussion

The authors might want to discuss that future studies would benefit from also collecting objective indices of mood (e.g. psychophysiological parameters) instead of only relying on self-report.

The authors highlight that participants were not required to perform any action to receive or lose points. To what extent might this have influenced the results? It is unclear why the authors took this approach that differs from previous research. Please justify and provide a rationale in the introduction.

Additional comments:

Table 1: …in these rounds is calculated (instea

6. PLOS authors have the option to publish the peer review history of their article (what does this mean?). If published, this will include your full peer review and any attached files.

Reviewer #1: No

Reviewer #2: No

---

## [Author Response · Author response to Decision Letter 0]

27 Jul 2023

We would like to note that we have addressed the concerns of the reviewers and the editor in our rebuttal letter which we have uploaded along with our revised manuscript.

---

## [Decision Letter · Decision Letter 1]

18 Aug 2023

An experimental approach to training mood for resilience

PONE-D-23-07979R1

Dear Prof. Stringaris,

We’re pleased to inform you that your manuscript has been judged scientifically suitable for publication and will be formally accepted for publication once it meets all outstanding technical requirements.

Thank you for providing the revised manuscript and your thourough response to the reviewers. All reviewers agreed to accept the revised version, which I fully support. One reviewer suggested a further emphasis on the sample (see copied text below). I think it is a useful comment and would recommend the inclusion of the "limited information of the sample". However, I leave this decision up to you, since you were indeed transparent about this issue and I evaluate this as rather minor limitation.

Kind regards,

Johanna Löchner

Academic Editor

PLOS ONE

Reviewers' comments:

Reviewer's Responses to Questions

**Comments to the Author**

1. If the authors have adequately addressed your comments raised in a previous round of review and you feel that this manuscript is now acceptable for publication, you may indicate that here to bypass the “Comments to the Author” section, enter your conflict of interest statement in the “Confidential to Editor” section, and submit your "Accept" recommendation.

Reviewer #1: (No Response)

Reviewer #2: All comments have been addressed

2. Is the manuscript technically sound, and do the data support the conclusions?

Reviewer #1: Yes

Reviewer #2: Yes

3. Has the statistical analysis been performed appropriately and rigorously? 

Reviewer #1: Yes

Reviewer #2: Yes

4. Have the authors made all data underlying the findings in their manuscript fully available?

Reviewer #1: Yes

Reviewer #2: Yes

5. Is the manuscript presented in an intelligible fashion and written in standard English?

Reviewer #1: Yes

Reviewer #2: Yes

6. Review Comments to the Author

Reviewer #1: I appreciate the time and effort put into improving this manuscript based on the reviewers' comments. I feel that this round of reviews has much improved an already good manuscript. (For future submissions: Personally, I would have preferred a document with tracked or highlighted changes, in order to be able to see at first glance what changes have been made since the first submission.)

I would like to recommend acceptance, but would encourage the authors to include some of the information provided in the rebuttal letter in the manuscript as well. The authors' responses to the reviewer comments provide valuable context to their results which are important not only to reviewers, but to all readers of their research. These points include: use of "approximately"; distribution of participants across tasks via convenience sampling rather than randomization and how it was ensured that each participant took part only once; considerations of power/sensitivity; considerations on different task responsiveness between participants and how this may relate to generalizability (which is limited not only because this is an online sample but because we have very little sociodemographic information about this sample and cannot judge how representative it is). This could be included in the manuscript either in the main text (as an additional sentence or a footnote where appropriate) or in the supplement. Without these inclusions, I expect many readers would be left with questions similar as those posed by the reviewers.

Reviewer #2: The authors were very responsive to all points raised by the Reviewers and provided a detailed rationale for their approach in their response. I have no further suggestions for improvement or concerns.

7. PLOS authors have the option to publish the peer review history of their article (what does this mean?). If published, this will include your full peer review and any attached files.

Reviewer #1: No

Reviewer #2: No

---

## [Editor Report · Acceptance letter]

23 Aug 2023

PONE-D-23-07979R1 

An experimental approach to training mood for resilience 

Dear Dr. Stringaris:

I'm pleased to inform you that your manuscript has been deemed suitable for publication in PLOS ONE. Congratulations! Your manuscript is now with our production department. 

Kind regards, 

on behalf of

JProf. Dr. Johanna Löchner 

Academic Editor

PLOS ONE